# A Theoretically Informed Critical Review of Research Applying the Concept of Liminality to Understand Experiences with Cancer: Implications for a New Oncological Agenda in Health Psychology

**DOI:** 10.3390/ijerph20115982

**Published:** 2023-05-29

**Authors:** Paul Stenner, Raffaele De Luca Picione

**Affiliations:** 1School of Psychology and Counselling, The Open University, Walton Hall, Milton Keynes MK7 6AA, UK; paul.stenner@open.ac.uk; 2Faculty of Law, Giustino Fortunato University, Via Delcogliano 12, 82100 Benevento, Italy

**Keywords:** oncology, chronic illness, liminality, threshold, crisis, processuality

## Abstract

Liminality was described more than 20 years ago as a major category explaining how cancer is experienced. Since then, it has been widely used in the field of oncology research, particularly by those using qualitative methods to study patient experience. This body of work has great potential to illuminate the subjective dimensions of life and death with cancer. However, the review also reveals a tendency for sporadic and opportunistic applications of the concept of liminality. Rather than being developed in a systematic way, liminality theory is being recurrently ‘re-discovered’ in relatively isolated studies, mostly within the realm of qualitative studies of ‘patient experience’. This limits the capacity of this approach to influence oncological theory and practice. In providing a theoretically informed critical review of liminality literature in the field of oncology, this paper proposes ways of systematizing liminality research in line with a processual ontology. In so doing, it argues for a closer engagement with the source theory and data, and with more recent liminality theory, and it sketches the broad epistemological consequences and applications.

## 1. Introduction

Liminality was described more than 20 years ago as a ‘major category of the experience of cancer illness’ [1]. Since then, it has been widely used in the field of oncology research, particularly by those using qualitative methods to study patient experience [2,3]. Through a critical engagement with this literature, this paper will show that a key strength of liminality theory is precisely the focus on the *experience* that it encourages. Essentially, the patient encounter with cancer (whether illness or treatment) is characterized in this literature as the *liminal experience*. This characterization has opened up many fresh insights into the challenges faced both by cancer patients and by those caring for them. For the most part, however, liminality theory is deployed in a sporadic manner in the literature. It lacks engagement with both the original sources and with more recent liminality work beyond the special area of oncology or, for that matter, the fields of health and medicine. This has limited the scientific growth potential of this field, giving the impression that liminality is recurrently re-discovered in relatively isolated empirical studies.

To address the problem just sketched, this paper takes seriously the need for theoretical critique and creative development. In the current historical conjuncture, health psychology is called to entertain new epistemological frameworks and models. To this end, the current paper confronts existing liminality research in oncology with more recent theoretical work which has proposed liminality as a fundamental ontological notion with broad epistemological consequences and applications [4,5,6,7]. In making this case, we draw attention to the embeddedness of liminality theory within a broader ontology of *process.* In the context of process thinking, liminality theory can help to deepen health psychology’s appreciation of its own embeddedness—and that of the patients it studies—in wider historical and societal processes [8,9]. It offers fresh ways to grasp the dynamics of liminal experience on multiple levels. It sheds light both on the crisis of sensemaking induced by cancer diagnosis (as a fracturing of semiotic borders leading to experiences of suspension and disorientation), and on the changing contexts in which such an experience unfolds. This helps to illuminate important issues of chronicity with implications for treatment and care and connects to liminality research beyond patient experience of cancer and its treatment.

The main section of the paper (Section 3) presents a theoretically informed critical review (TICR) of research which applies the concept of liminality within oncology beginning with the seminal work of Little et al. in 1998 [1]. Given our distinctly theoretical concerns, a formal qualitative evidence synthesis or QES was not appropriate. QES is the preferred term of the relevant Cochrane Group for a loose collection of procedures including qualitative systematic reviews, qualitative meta-syntheses and qualitative research syntheses [10]. These and other techniques are suitable for integrating findings from primary qualitative research studies at a higher level of generality, but they are less suited to the task of revealing how subtle differences in theoretical formulation shape and limit empirical foci and conclusions [11]. Further technical details about the TICR are provided in Section 3, and Section 2 below provides the background necessary for comprehending the theoretical stance informing the critical review. This includes an overview of the three main sources of liminality scholarship (Section 2.1), a discussion of liminal experience and the key problem of ‘stuck chronicity’ (Section 2.2), and an outline of liminality theory as a species of process thought (Section 2.3. Readers lacking the time and background necessary for scientific theory may skip directly to Section 3. A proper grasp of the process theoretical shift under consideration in the concluding section will, nevertheless, require full familiarity with the theoretical material in Section 2.

## 2. Theoretical Background

### 2.1. Three Sources of Liminality Scholarship

Liminality is not a common word, and its growing popularity within health research is perhaps surprising. The modern use of the term liminal (meaning *of the limit* or *threshold*) has three main sources, all quite abstract [7]. First, in 19th century psychophysics, the threshold of awareness was described as the *limen*. Only a stimulus capable of crossing this threshold could be transformed into a conscious report (hence, the notion of messages which operate in a ‘subliminal’ way). Second, in anthropology, the middle phase of rites of passage was named *liminal* (‘liminaire’), because during this phase, an emphasis was placed upon transition from one social world or state or status or station to another. Van Gennep [12] was interested in the fact that in practically all the societies that had been studied by anthropologists, the transition moment of these passages acquires a sacred status which requires them to be marked and managed by means of religious rituals. Third, the term *liminality* was a later arrival, used within Victor Turner’s processual ‘anthropology of experience’ as a distinctive modality of experience associated with, but not exclusive to, the liminal phase of rites of passage [13]. For Turner, ‘liminality’ is a particularly valuable mode of experience made possible by the temporary removal or suspension of the limits that usually shape, frame, or otherwise lend structure to human experience, activity and communication. For Turner, there is a parallel in this respect between the liminal experiences occasioned by religious ritual and what he called the *liminoid* experiences occasioned by art forms such as the theatre. Liminality is the experience of passage made possible when the limits that are usually in place (to structure experience, and hence, prevent passage) are suspended. In Turner’s account, liminal occasions generate fecund but disturbing conditions of ‘anti-structure’ that are associated with a shared sense of ‘we’ feeling or ‘communitas’ and the creation of novel identities, symbols and forms.

### 2.2. ‘Unstaged’ Liminality and the Problem of ‘Stuck’ Chronicity in Health Research

It was only in this third extended context of use (to describe an anti-structural mode of experience) that the concept of liminality was first applied to describe experiences associated with illness, and cancer in particular. This application to illness required an additional theoretical step, which had already been suggested by Turner. Namely, it required a distinction between two related and intertwined ‘types’ or perhaps meanings of liminality. First are what Turner called ‘staged’ liminal experiences created by the deliberate, temporary and symbolic removal of limits, such as what one finds during rituals and theatre performances. Second are ‘unstaged’ liminal experiences caused by unplanned and unexpected disruptions to ordinary life. Before he encountered van Gennep’s book, Turner had approached the latter using his framework of ‘social dramas’, but after the encounter, he took steps to unify them (as ‘staged’ and ‘unstaged’ ideal types) under a liminality theoretic framework. Both ‘types’ have the potential to engender or unleash unusually ‘unlimited’ experiences, but in unstaged cases, there is no reason to assume that this removal of limits will amount to a liminal experience of *passage*. Indeed, liminality research in the field of health has generally focused on the *absence of a sense of passage.* Liminality research within oncology in particular has got ‘stuck’ on the idea of a liminality which is ‘stuck’. Part of the reason it has got ‘stuck’ on ‘stuckness’ is that there has been insufficient acknowledgement of the fact that diagnoses with cancer provide clear examples of ‘unstaged’ experiences in which questions of troubled chronicity are highly likely to feature for obvious reasons.

Acknowledgement of a taken-for-granted application of the notion of ‘unstaged’ liminality to chronic health concerns would be relevant beyond the field of oncology and would explain why liminality research on health issues has concentrated predominantly around two central but distinct problems of chronicity. Namely, there is:A focal center associated with experiences of cancer;A focal center around conditions whose chronic nature is connected to the fact that their explanation remains unclear or problematic for some reason.

The type of chronicity typical of cancer is connected with the seriousness of the disease, which, though well understood and relatively easily diagnosed, may have no end. It may change the cancer patient permanently, alter their social status and communication possibilities and could end their life. Other diseases which share these features have also attracted liminality research, including MS [14] and cystic fibrosis [15]. Serious illnesses such as cancer, rightly or wrongly, raise the specter of whether recovery is ever possible, and in so doing, raise the possibility of death. The ‘stuck’ chronicity of chronic pain [16] and other largely unexplained or hard-to-explain conditions such as ME or CFS [17], by contrast, is connected to the fact that, unlike cancer, they are not well understood in a consensual way, and that this lack of understanding generates controversies that feed into experiences with these conditions. Experiences of those with chronic disabilities such as paraplegia or quadriplegia have also long attracted liminality theoretic research [18].

### 2.3. Outline of a Liminality Theoretic Framework as Part of a Process Ontology

Within the oncological domain, the notion of a distinctive type of experience called ‘liminal experience’ is routinely assumed but rarely explicated. Liminality is treated as if it were a naturally existing category and as if liminal experience could be assumed from an outside (third-person) perspective, described as an empirical fact and used within an explanatory model. For example, those faced with life-threatening illness are said to ‘descend into a liminal state’ [19], and this ‘state’ can then progress through various ‘stages’ [15]. Unless helped by health professionals and carers, those in this condition can fail to re-surface, unable to ‘exit the liminal state’ [20]. This tendency to naturalize neglects the fact that liminality is a *theoretical* concept designed to draw attention to experiences of transition ‘in process’ (i.e., as it is actually happening, from the perspective of those to whom it is happening). Liminality does not *explain*, but draws attention to occasions of becoming which are inherently ambiguous because ‘unformed’, and hence, as it were, ripe—not for ‘explaining’, but for ‘being explained’ (liminal experiences *call for* interpretation and motivate it). The primary value of liminality as a concept is that it recognizes an ontological reality to such occasions of becoming and transition that more familiar scientific ontologies deny or miscrecognize. It is for this reason that liminality research is applicable to so many subject matters at *all kinds* of scales, and that it ‘escapes’ delimitation, opening insights and forging connections between unexpected domains. Liminality, in sum, is not something *concrete* but the very process of *concrescence* (and its opposite).

The tendency to ‘essentialize’ liminal experience as something concretely empirical and natural, and with explanatory potential, it is traceable, as detailed in the following critical review, to the seminal influence of the excellent but theoretically rather partial contribution of Little et al. [1]. Under the influence of the brilliance of this publication, insufficient attention has been given to the fact that van Gennep quite deliberately did *not* include matters of illness, disease and health in his discussion of rites of passage. This certainly does not mean that a liminality theoretic framework is irrelevant to questions of health in general and oncology in particular. Rather, it means giving proper theoretical attention to the ways in which liminality theory inclines researchers to approach problems *processually*. An advantage of this more or less implicit ‘process ontology’, we suggest, is that it helps both the researcher and the people they study to recognize the reality of aspects of existence that tend to be ignored, disqualified or rendered invisible by the more familiar ‘substance ontology’ [21,22]. Crudely, substance ontologies assume that the ultimate realities are concrete entities in causal relationships, while process ontology challenges this assumption and give primacy to *activities*, i.e., to the ‘concrescent’ processes that actually give rise to empirical regularities [23]. The disadvantage is that this process ontology is conceptually demanding, and thus, there is a high likelihood that researchers will deploy the concept of liminality within some variant of the old substance ontology, mistaking it for one more causal variable in a positivistic explanatory model, or essentializing the liminal experience as if it were describable in a factual way from a third-person perspective (and not a theoretical lens inclining the researcher to take subjectivity more seriously). The result is often a theoretical ‘Frankenstein’s monster’ that, as it were, staggers a few empirical steps before collapsing under attack from its maker.

The important turn towards a process ontology is already implicit in the work of van Gennep, and was amplified by Turner (who referred to himself as a *process* anthropologist). At its heart, notions of *passage*, *transition* and *becoming* play the role of the phenomena that are most real (because they are *actual*): the basic ontological categories concern ‘actualizations’. From this perspective, more ‘static’ phenomena (states, substances, positions, structures) are secondary effects of emergent patterns within flowing activities. This means that key concepts such as *liminality* must never be mistaken as *states* or reduced to something *spatial* (though liminal experience obviously *involves* space). Van Gennep crafted the concept of the liminal as part of a groundbreaking work of anthropology in which he introduced to that field a new category of ritual ceremony: the *rites of passage*. For him, the term ‘liminal’ names the middle ritual phase of a three-fold processual pattern involving an initial phase of separation and concluding in a phase of reincorporation. Amid the vast cultural diversity described by anthropologists, van Gennep had discerned a universal *pattern* in which significant changes to people’s daily lives were marked by this pattern of ritual. He used the word ‘liminal’ (‘liminaire’) to describe a particular category of rite, which he also called a *transition* rite. Given an expected life change, liminal rites are enacted between separation rites and rites of reincorporation. Those involved have already symbolically *separated* from their prior form of psychosocial life but have not yet been *reincorporated* into a new form, newly recognized by others. As Turner would famously express it in his further development of ‘process anthropology’, they exist as if suspended in an unusual condition of being ‘betwixt and between’: no longer what they *were* but not yet what they *will be*. As further discussed below, this emphasis on temporality or—better—*chronicity* (from *Kronos,* which symbolizes the unity of past, present and future) has the potential to give a new vitality to the *chronic* nature of life experiences with cancer.

To adequately grasp this shift in ontology it is also necessary to emphasize the *symbolic* aspects introduced by liminality theory. Rites, rituals and ceremonies of all kinds, for example, are densely *symbolic* and this feature—which was central to van Gennep’s work—cannot be ignored. In particular, the symbolism typical of rites of passage is a processual symbolism entailing a flow of passage that occurs between a release from the form of a temporarily static state of psychosocial existence (pre-liminal ‘separation rites’), on the one hand, and a return to a new temporary ‘settlement’, on the other (post-liminal ‘reincorporation rites’). For example, separation rites often involve a symbolism of *cutting*, such as when hair, nails or a cord is cut or an adolescent is circumcised during an initiation rite. Separation rites also often utilize space symbolically, as when initiates or pregnant women are removed to a separate dwelling on the edge of a village (van Gennep discusses in this spatial respect an entire category of ‘territorial rites’). Reincorporation rites often involve a symbolism of *tying* or *unity* as with the wearing of a ring, the tying of knots, enthronement, or the sharing of a reunion meal. The symbolism of liminal rites tends to evoke *movement,* such as passage through a portal, carriage across a threshold, wandering in strange territory or diving from high points. When engaging in ritual ceremonies people are, as it were, immersed in what Turner called a subjunctive realm of a symbolically mediated *as if* experience that has been carefully distinguished from the more mundane *as is* experiences of ‘ordinary life’ as it unfolds under the guise of one of its temporary settlements.

This emphasis on symbolism should in no way imply that the transformations marked and evoked by rites of passage are not *actual*. In fact, the *as is* pregnancies, births, deaths, betrothals, marriages, territorial movements and seasonal changes ‘managed’ by *as if* rites of passage are as actual as any life events, and often rank among the most important. Rather, the emphasis on symbolism points more fundamentally to the (ever-changing) symbolically mediated basis of human psychology and society *as such* (a perspective which resonates with Sontag’s [24] famous orientation to the role of metaphors in experiences with cancer). The rites of passage indicate how highly ‘staged’ symbolic forms of human sensemaking are designed to intersect with the shifting flow of more mundane and material life circumstances. The ritual enactment of a purely symbolic separation, transition and re-establishment is, thus, *layered upon an actual separation, transition and re-establishment*, and the former can give the latter a social and personal meaning they might otherwise lack (in the case of the anthropological data dealt with by van Gennep, a *sacred* meaning). Nevertheless, this intertwining and intersection of ritual and life, ‘staged’ and ‘unstaged’, raises the difference between what might better be called *devised* liminal experiences that are deliberately occasioned by symbolic forms, such as ritual (but also aesthetic symbolic forms, such as theatre and music, and ludic aesthetic forms, such as sports and other games) and *spontaneous* liminal experiences that are occasioned by unexpected events—such as a diagnosis of cancer—that disrupt the existing forms of sensemaking (that lend structure to lives) and force a change to practices and sensemaking that could previously be taken for granted (see Stenner [7] for this critique and redeployment of Turner’s ‘staged/unstaged’ distinction). Ritual, from this perspective, can be viewed as a distinctive *symbolic medium* with ancient roots that long predate, for example, the invention of writing. However, ritual continues to be relevant within social systems that have also developed other symbol-mediating technologies, such as writing, printing, computers and the internet, and that operate with more abstract symbolic forms, such as science and modern medicine [25]. The point here is not that rituals are symbolic but ordinary life is not. On the contrary, it is precisely because all human life (from the settled to the liminal) is symbolically mediated that it becomes important to distinguish between those occasions when symbolism is florid and explicit (during rituals, when engaging with the arts, when playing sport, etc.) and those occasions where it sinks to invisibility and passes for the real, quotidian, world.

## 3. A Theoretically Informed Critical Review of the State-of-the-Art Use of Liminality Theory in the Oncological Context and Beyond

### 3.1. The Literature Search

In this section, we critically discuss a selection of articles published between 1998 and 2023 on the topic of liminality in the context of cancer and other chronic health conditions. The search terms ‘liminality AND cancer” and “liminality AND chronicity” were entered into PubMed, SCOPUS and Google Scholar and additional publications were sourced by consulting the reference sections of key papers. Of the resulting 411 publications, 112 were discarded as redundant and 181 were not pertinent to the criteria. The remaining 118 publications were organized by the second author (and confirmed by the first author) into four groupings: liminality used as an interpretive lens to categorize study results; liminality as a conceptual perspective orienting the aim of the research; liminality as a notion with presumed relevance for methodology and choice of research tools. Examples of publications representative of each are discussed below. As further discussed, many publications use the words ‘liminality/liminal’ as a synonym for ‘confusion’, ‘distress’, ‘disorientation’, etc. without showing critical awareness of the distinctive meaning. We also observe that liminality research has burgeoned in several other fields, especially where ‘boundary processes’ are considered fundamental to scenes of transformation or ‘becoming’ [7,9,21,26,27,28,29,30], including research on lifespan transitions, such as migration, mourning, new parenthood, changes of residence or career, passage from school to college, etc. [31,32,33,34,35,36,37,38].

### 3.2. Seminal Contributions of Frankenberg [39] and Little et al. [1]

It is important to note an early use of liminality in the field of health research prior to 1998 by Frankenberg [39]. Frankenberg followed Turner in approaching sickness in general as symbolically mediated by ‘cultural performances’ enacted under conditions in which the usual routines that structure lives are suspended by a disruption of the normal order. This research set the pattern of considering serious illness as ‘liminal’ in the sense that it is associated with the collapse of the normal order of things taken for granted by the sick person. In doing so, Frankenberg carved out a place for the introduction of socio-cultural and psychological factors into a terrain otherwise dominated by the bio-chemically oriented positivism of medical science. The concept of liminality, thus, allowed some credence to be given to realities that would otherwise be excluded from consideration by the framework supplied by the dominant science (a kind of ‘making social’ of disease). It allowed attention to the role of metaphor under conditions of challenge to sensemaking, for example. There is a reflexive aspect of this development. Just as a cancer diagnosis can be understood as a ‘newcomer’ which disrupts an existing flow of an individual’s (and family’s) life-practice and sensemaking, the advent of a liminality-theoretic approach within the health field can be understood historically as a ‘newcomer’ which disrupted (and demanded a re-formation) of the existing flow of medical scholarship. In this sense, liminality scholarship is itself ‘liminal’ to the extent that it provokes a transition. Indeed, on the basis of such insights, by the 1990s, a ‘narrative turn’ was establishing itself in a variety of disciplines, including the fringes of the fast-developing field of health psychology [40]. This served to consolidate those theoretical perspectives ‘critical’ of the positivistic disregard for subjectivity, and to champion qualitative methods as ways of taking ordinary discourse and experience seriously [41]. In studies of health issues, the former manifested as a critique of the ‘medical model’ and its tendency to exclude the relevance of psychosocial factors, and the latter as a program of research into ‘patient narratives’ [42]. It was in this broad context that liminality theoretic accounts of people’s experiences with cancer were first articulated.

The watershed contribution from 1998 by Little et al. [1] made a lucid case that liminality be recognized as a ‘major category of the experience of cancer illness’. They drew attention to three recurring features in the experiences described by the 10 patients they interviewed about the outcomes of their colon cancer treatment. First, they noted both the *immediate and profound impact* of the cancer diagnosis on the patient’s life pattern and the seemingly *permanent* changes this brings to the patient’s identity (they remain a ‘cancer patient’ regardless of time since treatment or (non-) recurrence of the disease). Second, they attend to the impacts of this change on the quality of social interaction and describe a quality of lasting ‘alienation’ from others. As if thrust suddenly into a different ‘world’ with a new ‘identity’, their participants report difficulties in communicating the nature of their experiences to those they were previously close to. Third, participants report a persistent heightened sense of their existential *boundedness*, experiencing a highly limited sense of time, space and their own powers of activity.

These three findings were, however, self-consciously simplified. This process of complexity reduction was guided by a particular interpretation of the framework of liminality theory. Little et al. [1] (p. 1485) were admirably clear about this, stating that these subjectivities were experienced ‘in varying degrees’ and that ‘individual responses to these experiences were complex and variable’. The value of the category of *liminal experience* for Little et al. is that it allows the researcher to understand all three findings in a unified manner. In short, all are aspects of a ‘state’ of liminality that arises following the life disruption occasioned by the cancer diagnosis. The three aspects are part of a process of change that pertains, not just to the organic body, but through that body to the entire life of the patient. This gives an interpretive power to the ‘category of liminality’ that goes beyond the theory of ‘biographical disruption’ that by 1998 was also already well established [43,44]. For example, it seems clear that the corollary to van Gennep’s phase of *separation rites* is Little et al.’s [1] (p. 1485) identification of an ‘initial acute phase of liminality’ characterized by disorientation following diagnosis. This is the phase of abrupt separation from the life that had previously been taken for granted. Terms such as ‘acute’ and ‘abrupt’ here mark a distinctive time experience that characterizes this phase for many. Patient’s report how they must suddenly ‘drop everything’ and seek treatment as a matter of urgency, for example. However, the parallels stop there. In fact, if one compares to the classic accounts of rites of passage discussed above the main findings of Little et al. [1], it is clear that they are antithetical to what would be expected from liminality theory. Hence, in place of a liminal phase followed by a phase of reincorporation, one finds the initial ‘acute’ phase followed by just one further phase: an ‘adaptive, enduring phase of suspended liminality’. Instead of a passage to the new beginning of a new form of life, the patients described a sense of being stuck in an enduring state of suspension: a paradox of permanent liminality. Furthermore, instead of the rich and magical communicative bonds that Turner associates with the sense of *communitas* generated through shared experience of liminality, one finds a sense of ‘alienation’ from others with whom one can no longer communicate. Moreover, in place of the liminal experience of unlimited ‘super-human’ horizons connecting one to cosmic processes on an eternal time scale one finds experiences of limited time and constrained powers.

Little et al. are clearly aware of these discrepancies. They handle them through the argument that their notion of liminality is distinct from that of van Gennep or Turner. ‘Our liminality’, they state, ‘is an enduring and variable state’. The trigger for ‘entry’ into the liminal state is essentially the ‘labelling inherent in the cancer diagnosis’. Their distinctive notion of liminality, however, is not discussed in depth, and in making this move, they risk separation from the very tradition that spawned their key concept. Reference is made instead to the tradition of existential philosophy the premise of which, allegedly, is that ‘liminality is the mode of life in which we must live’. For Little et al., liminality is the rise to explicit consciousness of a usually implicit (and carefully avoided) state of dread that is triggered by a diagnosis of serious illness which interrupts life processes (organic, psychological, collective) and forces a confrontation with the horrific *nothingness* of reality. Poisoned by the nothing, we find ourselves—similar to Tolstoy’s Ivan Illich—in a different world, no longer able to communicate with those more able successfully to eliminate ‘liminality’ from their lives.

Little et al.’s ‘existentialization’ of liminality theory, while interesting and groundbreaking, is somewhat problematic for three reasons. First, because the tradition of thought from Kierkegaard to Heidegger, Sartre, etc. does not in fact use the term liminality. Second, because the fundamental concepts this tradition does articulate center on the ‘gloomy challenge’ of anxiety and dread in the face of death and, later, the ultimate absurdity and meaninglessness of existence. Indeed, a little reflexivity here would show that the gloomy existentialism of Sartre (a one-time supporter of Stalin) and Heidegger (a one-time supporter of Hitler) itself arose during the liminal phase between two massively destructive world wars which drained Europe of its sense of purpose and value. Arguably, nothing could be further from van Gennep’s emphasis on rites of passage as events of symbolic *rebirth*, which, if anything, replenish the relationship between the individual, collective and cosmos [23]. Third, paradoxically, because it universalizes liminality as a quasi-natural ‘state’: ‘We believe that the state into which the survivor of serious illness, or the person with chronic illness, enters is one of liminality, and that this state persists in some form or other for the rest of the patient’s life.’

This third weakness is important because, as noted in our introduction, ‘liminal experience’ is always subject to dynamics of culturally mediated *interpretation* by means of symbolic forms of different kinds. We have shown why this fact is not accidental. Liminal experience ‘as such’ is precisely ambivalent, ambiguous and paradoxical: both/and *as well as* neither/nor. It is neither one thing nor the other because it is an experience that hovers in a subjunctive condition of potentiality, open to actualization in multiple ways. As such, it does not itself ‘explain’ anything but ‘calls out’ as it were, for interpretation. It can be interpreted as sacred experience through the mediation of religious symbolism (as van Gennep illustrated), or as aesthetic experience through the mediation of artistic symbolism (as Turner illustrated) or as ludic experience through the mediation of the symbolism of play and sport (as Stenner discussed [7]). Little et al. have interpreted the liminal experience of their patients through the symbolic form of existential philosophy, and by means of this form, they treat liminality as if it were a known, yet disturbing, quantity. However, through this interpretation, they collapse and ‘de-paradoxify’ the inherent potentiality of liminality onto just one of its possible values: it is dread and angst (*but not* excitement and stimulation). However, this collapse of the equipotentiality proper to liminality is a dangerous step because it risks denying its own interpretative effects and projecting them onto the patients in their study. Moreover, what it leaves out is precisely the constructive and creative focus introduced by van Gennep and Turner, whereby the symbolic elaboration of liminal experience is geared towards the problem of *becoming other*. Worse, it recruits all questions of becoming to the existentialist theme of the authentic being-towards-death: above all, one must have the resolve to become ‘authentic’. Put differently, all questions of becoming other must be wrestled from the tyranny of ‘other people’ and be freely chosen by an authentic individual. All questions of meaning, in short, are to be freed from older and redundant symbolic forms and decided by the newly born authentic existential subject.

Of course, as hinted above, there is no reason to assume that *spontaneous* experiences of serious illness *should* entail productive dynamics of becoming other, and that is precisely why van Gennep did not include them in his survey of rites of passage. Again, this does not mean that serious illness (or spontaneous liminal experiences in general) is irrelevant to liminality theory. It certainly is relevant, but it must first be acknowledged that liminality theory primarily concerns passages of becoming that give rise to what can symbolically be called re-births. To give some examples, in initiation rites, the symbolic death of the child is the symbolic birth of the adult. In marriage rites, the symbolic death of the ‘single’ adults is the symbolic birth of the married couple. In funeral rites, the symbolic death of the deceased is the symbolic birth of a new phase of their spiritual existence beyond the body. In seasonal rites, the symbolic death of winter is the symbolic birth of spring, etc. A spontaneous experience of a weakened or destroyed life is not yet a liminal experience unless and until it begins to give rise to new forms of symbolism which nourish new modes of livable life, it is at best the paradox of a liminality become permanent: a ‘liminal hotspot’ [9]. In short, this existential interpretation naturalizes a one-sided construction of liminality which conceals the decision to focus on *spontaneous* experiences to the neglect of the *devised* experiences that were the main interest of van Gennep and Turner.

### 3.3. Extensions and Critiques of the ‘Existentialization’ of Liminality

Since Little et al. [1], many in the oncological field have interpreted narratives derived from interviews with patients in terms of liminal experiences of disruption at organic, psychological and socio-communicative levels of process [14,15,16,17,18,19,20,21,22,23,24,25,26,27,28,29,30,31,32,33,34,35,36,37,38,39,40,41,42,43,44,45]. The disruptive and traumatic impact of diagnosis on identity in particular has been regularly described as a distressing sense of disorientation, fragmentation and uncertainty which gives rise to liminal experience [46,47,48]. Many too have reported findings consistent with their discovery of a subsequent sense of ‘alienation’ from others, including difficulties in communicating the nature of their experiences to those who have not been through something similar. Thompson [47], for example, reported two forms of ‘communicative alienation’ among nine female interviewees living with stage-three ovarian cancer corresponding to difficulties *articulating* their own experiences due to the emotional and physical impact of treatment (‘I lost my words’, ‘I couldn’t express myself’), and difficulties *sharing* their experiences with those close to them (they have to ‘deal with everyone else’s reaction’ and find that regardless of whether you *are* ok, ‘people want you to be ok’). The unexpected ‘newcomer’ (both the cancer and its treatment and its impact on significant others) evades communication in both of these senses because it disrupts daily life, including emotional and relational ties, and normal work and leisure activities, introducing new paradoxical conditions of communication.

Much of this literature continues Little et al.’s ‘existentialization’ of liminality, which is also further developed—in the form of a differentiation between ‘mortality salience’ and ‘death salience’ [49]. For example, Frommer [50], in an article on ‘liminal spaces of mortality’, urged that ‘our mortality… defines the human condition’, and Dunn et al. [51] described the experience of patients undergoing allogenic stem cell transplantation for hematological malignancy as an ‘existential crisis’ that follows from an encounter with ‘their own mortality’. In a meta-synthesis of Nordic studies on the patient experience of intensive care, Egerod et al. [19] identified as their overarching theme: ‘the patient experience when existence itself is at stake’. Willig and Wirth [52] connected Heideggerian ‘death awareness’ to ‘terror management theory’. We do not doubt the profound truth of this proposition, but the danger is that it limits the concept of liminality to an inner experience located, as it were, ‘inside’ the individual traumatized by facing their own death: a ‘liminal *state’*. An extreme example of this is given in a study of liminality and head and neck cancer by Dawson et al. [53], who stated: ‘we identified liminality *in* people who had received surgical treatment of head and neck cancer’ (we have added the italics). This might seem an innocent assumption, but caution is necessary. Consider, for example, the observation made by one of Thompson’s [47] interviewees: ‘My husband really feels that I woke up from the surgery and, he said, “I’ve lost my wife.” He just even doesn’t know who I am anymore. But it’s like, how can I not change?’

This data extract makes clear that the liminality in question does not simply ‘belong’ to the person with cancer—in this case the woman speaking—as a ‘state’ of their traumatized consciousness existing ‘in’ them. Rather, in this case, the *husband* is the one unable to ‘process’ the ‘loss’ of his wife. However, it is not he who is dying, and—quite importantly—she is also not (yet) dead. The liminality here refuses, as it were, to be contained within the limits of a given psyche, and travels between this husband and wife: her liminal experience (of not being able to communicate with him, for example) is intimately connected to and conditioned by his liminal experience (he makes communication with her paradoxical because, among other things ‘he doesn’t even know who I am anymore’). This could be grasped by observing that liminality is not static but precisely *travels* between his experience and hers, affecting both [32]. What appears one moment to be a state of mind of the one facing their own inauthentically repressed being-towards-death, morphs before our eyes into a mobile dialogical property of a perturbed dyad. McKenzie and Crouch [54] have also pinpointed this predicament, noting in italics: ‘*Perhaps it is the case, then, that both cancer survivors and the persons close to them want responses from one another which are quite different from those they are actually getting*’. These authors traced this problem to a ‘mismatch’ between the ‘interaction order’ and ‘individual psychology’ that leads to ‘discordant feelings’, but we must be careful to recognize that (a) any ‘discordant feelings’ do not simply belong to the person with cancer and (b) ultimately, no firm distinction can be drawn between the communication-oriented ‘interaction order’ and the subjectivity-oriented ‘individual psychology’.

Thompson’s [47] article is worth dwelling on because she illustrated a number of the points we wish to make. First, similar to Little et al. [1], Thompson was impressed by the integrative potential of liminality theory. Her grounded theory analysis yielded seven categories of experience (trauma, loss, suffering, death anxiety, social supports, spirituality and narratives of change), but she concludes that each one ‘related to one central category—liminality, initially defined in the terms set out by Little et al. Nevertheless, Thompson identified a difference between the liminality experienced by *her* participants and by Little et al.’s participants. Here, it seems, liminality is morphing and ‘traveling’ again, for clearly what is at stake here is not simply the *participant’s* conception of liminality (since to our knowledge, none of these participants use that word, because—after all—it is a theoretical instrument of the observer) but that of Thompson [47] and Little et al. [1]. For now, however, we merely note that Thomposon found it necessary to ‘broaden’ the latter’s definition in two ways: to accommodate the ‘social surround’ of liminal experience and to appreciate its ‘generative potential’.

### 3.4. Broadening to Include the ‘Generative Potential’ Associated with Liminal Experience

The notion of ‘generative potential’ leads us to the heart of the problems surrounding Little et al.’s existential notion of an ‘enduring phase of suspended liminality’ which ‘collapses’ liminality to just one of its ambivalent potentials (the dysphoric). As noted above, van Gennep did not include illness rituals within his *Rites of passage,* doubtless because it is not clear that recovery from illness is a transformative matter of becoming other and gaining social recognition for a new becoming. Serious illness is more similar to a ‘crisis’ in that the medical interventions it prompts seek to facilitate a return to health following a ‘separation’ from health [54]. Restoration is not the same as the reincorporation of a transformed condition. Why, then, call the rupture introduced by illness a liminal transition if it can be adequately grasped simply as a crisis? (In fact, the Greek origins of the word crisis lie in medicine and law). This is a genuine question, not a rhetorical one that assumes the answer to be ‘there is no good reason to call serious illness liminal’. Much research has in fact indicated a relationship between serious illness and psychosocial transformation [17,51,55,56,57,58,59]. Thompson’s [47] ‘generative potential’, likewise, does imply the potential to become other through the passage of illness. However—importantly—nothing requires us to assume that the becoming must entail the end of the cancer. As she concludes:

‘having a catastrophic illness, such as ovarian cancer, may lead to a subjective sense of personality change, of deepening of ties to others and the transcendent, as well as an ability to pay exquisite attention to the nuances of daily existence. These changes were reported as being positive for the participants, shifts they directly attributed to having ovarian cancer and becoming aware of the finitude of life. This dynamic demonstrates growth within a liminal experience’ [47] (p. 346).

As well as restoring lost ambivalence to the theme of ‘alienated communication’, Thompson’s [47] version of liminality also questions the way in which Little et al.’s [1] version reduces their third theme of limited ‘boundedness’ to its negative potential. For Thompson’s participants, the sense of having limited time was connected with a transformed time consciousness that provided them with a sharper sense of the values of the truly important things in life: to no longer waste a time become precious. What for Little et al. were limitations in the sense of reductions in powers, for Thompson could be subjective *expansions* that can be used to ‘catapult an individual in a positive direction towards change’ [47] (p. 346).

Clearly, there is also reason to be cautious about such claims for ‘catapulting’ becoming or ‘growth’. Scully [3], for example, warned that ‘reframing liminality as a transformative opportunity would be misleading and offensive if it were interpreted to mean that patients just need to bring a better attitude to serious long-term illness’. We agree, but the crucial clause here is ‘if it were interpreted’ as a mere switch of ‘attitude’. As Thompson’s research makes clear, more than ‘attitude’ is at play when a cancer patient’s sense of identity is negated by their own loved ones. The key point is that once the assumption of a quasi-universal and basically negative ‘liminal state’ is questioned, it is possible to recognize that a diagnosis of cancer does not in itself (as if naturally, and universally) equate to a life of suspended liminality characterized by a permanent identity as a cancer patient, poor communication and life limitations. Rather, at least some of this negativity is attributable to a lack of recognition on the part of significant others, and hence, the discordant or dissonant attunement between self and others that Greco and Stenner [9] identify as a feature of a ‘liminal hotspot’. Thompson [47] astutely pointed out how Little et al.’s [1] discussion of communicative isolation does not fit a swathe of positive communicative experiences reported by her participants—‘Amazing experiences… incredible experiences of support’—when they met other women who had been through similar experiences with cancer. They described forming deeply intense and transformative relationships in support groups and in camps and online forums. These experiences do indeed evoke Turner’s descriptions of lasting *communitas* forged between liminars. It is not ‘liminality’ as such, but the discordant subjectivity of the interactional space that blocks the possibility of becoming by denying any recognition to emergent potentials. It is not the content of communication that matters here, but the quality of its ongoing dialogical form. An interlocutor who, failing to listen, insists—for their own reasons of liminality—upon expecting their partner with cancer to ‘grow’ from the experience, would not merely be offensive: they might effectively prevent becoming. Moreover, importantly, what cannot be assumed in advance or dictated from outside is any answer to the question: ‘becoming what’? In this context, we note McKenzie and Crouch’s [54] hesitation to accept Little et al.’s [1] two phase ‘model’ (acute and suspended) and to suggest a third stage which might take 5 or so years after diagnosis, in which sustained liminality is ‘transcended’ thanks to a returning sense of living more securely and similar to ‘ordinary people’. Again, however, we warn against imposing such a ‘three-stage model’ as if it were a truth abstractable from the realities of the actual lives that were, after all, the source of the abstraction.

### 3.5. Broadening liminal Experience to Accommodate Ever-Expanding Circles of the Social Surround

The second ‘broadening’ called for by Thompson [47] follows directly from the above. Once we step back from the existential tendency to naturalize and universalize liminal experience, it becomes necessary to recognize how the unfolding of that experience is always patterned and flavored by its ‘social surround’, from the micro setting to the macro. We have seen, for example, how the reactions and shocked expectations of carers can turn a liminal experience into a liminal hotspot, full of paradox and frustration (the ‘neither/nor’ aspect comes to predominate over the ‘both/and’). From this perspective, even the early stage of so-called ‘acute liminality’ (which might be better understood as a predominantly negative *crisis* phase), when the patient hears the news, ‘context’ matters, including the manner of the delivery of the news [59,60,61]. The acute crisis phase is likely to present as a ‘black box’, whose extent and intensity depends on the patient’s biography and resources, the nature and strength of their support groups, the nature and modalities of the medical news, the impact and output of treatments, etc. These things in turn will depend upon the specific characteristics of the newcomer producing the emergency (what kind of cancer it is, how advanced, etc.).

We have observed the tendency of liminality theory to ‘travel’, drawing the researcher into reflexive jumps to different levels of reality no less ripe for comprehension as liminal phenomena. This is directly related to Thompson’s [47] problem of ‘social surround’, since each jump accommodates more of the surround. We have seen, for example, how the liminal experience of a woman with ovarian cancer can turn out also to be the liminal experience of her husband, and this shifts our gaze from the individual to the couple as the affected unit. Indeed, liminality research has also clustered around the liminal experience of carers. Gibbons et al. [61], for example, produced an integrative review of 26 studies of family caregiving, and they concluded that a liminality frame is required to grasp the role ambiguity of family carers. However, patient and caregiver interaction is also itself nested in the communication processes and practices of a medical system, and liminality theory is no less applicable to medical systems going through crises and transformations. Willis et al. [62], for example, studied patients with terminal cancer and their carers in the context of a state-run health organization undergoing privatization. Here, the experience of the health-care professionals too was described as a ‘liminal journey’, but one which they took care to conceal from those in their care (a predicament which added to the liminality of their experience).

### 3.6. Recognizing That the ‘Societal Setting’ Is Itself Ever-Changing and That Liminal Experience Is a Generative Part of That Dynamic of Change and Continuity

In drawing attention to *becoming,* liminality theory foregrounds the relevance of psychosocial change and also of continuity within change. What Bury [44] called ‘biographical disruption’ is a spontaneous experience of rapid and usually unwelcome change. When the ‘structures of everyday life and the forms of knowledge which underpin them are disrupted’ [44] (p. 169), people face a critical rupture which violently *separates* them from a life that has suddenly become their ‘old life’. The sense of what is taken for granted about order and normality is distorted to the point of losing the perception of self in terms of continuity [63,64,65]. As Salvatore and Venuleo [66] and Greco and Stenner [9] argued when articulating the notion of a ‘liminal hotspot’, this includes transformed relations to self no less than to others, and it impacts time consciousness. What was the present abruptly becomes a past present. The future that belonged to that past present becomes a past future. The past that belonged to that past present likewise becomes a ‘past past’. From this perspective, liminal experience is a matter of living in a transformed present disconnected from the past and awaiting a ‘future future’. Furthermore, the paradox of permanently enduring or ‘suspended’ liminality shows up in this light as a matter of being stuck in the ‘hotspot’ of this existential disconnection. Expressed spatially, cancer survivors can feel themselves *on the margins* of everyday life, limited to a borderline condition between good and evil, survival and threat, experimenting with the borders of their experience that are never too reliable [52].

Long ago, Wheelwright [67] observed the peculiar time-consciousness of liminal experience, referring to it as an “atemporal present” [68] (p. 1), which, in essence, exists out of time. This itself can be paradoxical, as if everything happens everywhere and all at once, and yet, at the same time, nothing happens. It is this experience that underlies the alienation both from self and others, since experiences and activities can no longer be timed and spaced according to a familiar and shared pattern. The so-called ‘exit’ from liminal experience is a matter of reconnecting with one’s own time (securing a continuity with one’s past and a sense of future) and that of others. This territory remains poorly understood. Blows and colleagues [48] spoke of a condition in which recurrent “interpersonal emotional dissonance” leaves patients permanently “marked by the disease”. On the other hand, survivors run the risk of becoming one’s own *emotional carers*, investing and engaging in psychological effort to modify/alter/conceal their feelings and stay positive, control negativity and appear ‘normal’ in the effort to maintain close relationships and protect those close to them. However, due to the uncertainty created by the risk and the fear of recurrence, even many years later, it can be difficult to sustain constantly a positive attitude [54].

When McKenzie and Crouch [54] referred to how cancer survivors can live in a world of “dissonant interactions” where they cannot express how they really feel, they, and others, attributed this to society’s emphasis on warding off the specter of death and focusing instead upon positivity, strength and uplifting cancer stories [52,69,70,71,72]. In fact, this observation about the relevance to liminal experience of historical changes within society was an important theme in Little et al.’s seminal 1998 article [1]. Their argument, in essence, was that before the advent of techno-scientific modernity and—in particular—biomedicine, people were obliged to live with ever-present liminality because disease and death were ever-present fixtures. Today, by contrast, liminal experience has been, if not entirely eliminated, then at least pushed to the margins of social life and treated as taboo. The rise in the cultural relevance of cancer and its taboo as a ‘killer’ is connected to the fact that large proportions of the populations of wealthy capitalist societies are no longer killed or incapacitated by a host of other conditions and causes of death and injury, and can expect to live long and healthy lives [52]. When death and suffering are pushed to the margins, conditions such as cancer come—under these specific historical conditions—to ‘carry’ the disowned and disavowed liminality. There is, therefore, a paradox at play in Little et al.’s [1] proclamation that ‘liminality is the mode of life in which we must live’. Namely, it appears, by their own logic, to be the mode of life in which ‘moderns’ precisely *do not live* (until…). It is life without liminality that must be prolonged by all means for as long as possible. This historical story is, however, partial. It must be reconciled with its twin (versions of which have been articulated by thinkers as diverse as Deleuze and Guattari, Victor Turner, Rene Girard and Arpad Szakolczai), which holds that, on the contrary, so-called ‘pre-modern’ societies went to extreme lengths to contain and limit liminal experiences, particularly within the frame of the sacred, while modern capitalism is unique in ‘unleashing’ unconstrained and permanent liminality.

### 3.7. Liminality and the Use of the Arts and Ritual to Rebuild Ruptured Life Worlds

The distinction between ‘spontaneous’ and ‘devised’ liminal occasions introduced above draws attention to the difference between liminal experiences prompted by real-life crises (such as a cancer diagnoses) and liminal experiences deliberately created by means of culturally devised media (such as a movie about living with cancer). According to Stenner [7], ‘media’ such as rituals, tests of endurance, sporting contests, theatre, music and the arts more generally serve as ‘liminal affective technologies’ (LATs), functioning to shape and modify liminal experience. However, the true relevance of this distinction is to draw attention to how the experiences created these two ‘types’ come together [7,23]. Namely, experiences ‘devised’ by means of LATs can help people to make new sense of their experience as they go through crises and liminal becomings in their actual lives, and to work out escape routes from sticky liminal hotpots. Several recent studies have devised and utilized rituals as ‘liminal technologies’ during and after the disease experience [15,20,62,69]. Sleight [20], for example, developed a framework for the use of ritual as part of a psychosocial treatment approach to help cancer survivors move beyond ‘prolonged liminality’. Gibbons et al. [61] proposed reframing family caregiving as a rite of passage to facilitate creation of an environment conducive to the reconstructing of shared sensemaking. Others have used art forms of various kinds as liminal technologies for the management of cancer treatment and rehabilitation [2,14,19,52,62,70,73]. Lit [74] approached therapy groups for terminal cancer patients as opportunities for creating a ‘transformative space’ for the sharing of narratives, using an arts-based technique they called ‘double listening’. Charmaz [63] proposed a methodology based on encouraging ‘story telling’ among chronically ill people in order to attend to the silences which guard hotspots of liminality. Sibbett [75] proposed ‘arts-based auto-ethnography’ as a vehicle for making sense of the ‘limbo’ of their cancer experience. Roessler and colleagues [76] used liminality theory to illuminate how painting and music can positively shape the feelings of patients in cancer rehabilitation programs. Art therapy, from this perspective, operates as a liminal technology when it presents opportunities to express without words some of the problems patients struggle with [77,78].

### 3.8. Both/and and Neither/Nor as a Main Feature of the Liminal Experience

Much of the recent liminality-theoretic literature in the oncology field agrees upon the broad notion that liminal experience is deeply paradoxical such that it escapes an Aristotelian identity logic of being categorically ‘either’ this ‘or’ that, and generates forms of experience that are not just ‘both/and’ but both/and *as well as* neither/nor [38]. One recurring observation here is that the uncertainty caused by the possibility of cancer recurrence (as a probable/uncertain occurrence) means that individuals feel trapped between two psychosocial dimensions: health and disease [1,14,16,51,61,62,63,64,65,66,67,68,69,70,71,79]. Comparable characterizations regularly feature beyond cancer research in studies portraying people with chronic conditions, physical disability, schizophrenia and arthritis, who described feeling “*neither sick nor healthy*” [16,63,72,80]. We find the use of the notion also in those conditions of permanence in a comatose state, or of Alzheimer’s patients and the elderly as beings “*neither dead nor alive*” [80,81,82]. Murphy et al. [16] have used the liminality to describe the social life of people with chronic disabilities (e.g., paraplegia or tetraplegia) as a state in which they define themselves as “blurred and indeterminate” [16]. Studies on the health conditions of subjects have corroborated this assumption, showing, for example, that being at risk of developing cancer is understood as a state of “unhealth” [83], and “neither...nor”, “in betwixt and in between” [13,65,84,85]. In the case of Navon and Morag’s research [46] conducted on the narratives of patients at the end of prostate cancer therapies, referring to their different life contexts, they described themselves as ‘*emotionally disabled*’, ‘*a car without an engine*’ and ‘a *man who is not a man*’, and living ‘*together alone*’ with their partners. These subjects highlight the fatigue of searching, the difficulty of classifying themselves into culturally available categories that normally produce the sense of identity [79]. A change is triggered in one’s own psychosocial categories of identity: the subjects feel separate but in a relationship, differentiated but integrated, different but recognizable. In the absence of an internal and social scheme for managing their incompatibility with familiar categories of meaning, in the condition of liminality, the person goes through a phase of loss of identity, she considers herself as “*neither this nor that, but both*” [13].

Finally, at the root of this characterization lies the *temporal* dimension of becoming. Once we grasp that the notion of liminality implies transition or passage, then it becomes clear that liminal experience is what becomes of experience when its experiencer is no longer what they were but not yet what they will become. This makes the phase of transition inherently paradoxical, because the ‘passenger’ becomes a fluid mixture, one moment describable as *both* what they were *and* what they will be and yet *neither* what they were *nor* what they will be. This mixture, in short, is proper to a present whose future is a future future and whose past is a past past. In devised experiences, this paradox can be welcomed and accepted, and people with experience can guide the passenger through the process without the paradoxes becoming vicious circles of entrapment. In spontaneous experiences such as the traumatic experience of a cancer diagnosis or when going through its medical therapeutic process, there can be less ‘tolerance’ of the paradoxes, and a tendency to force them into simplified Aristotelian categories (‘de-paradoxification’). This explains why liminal experience can be both the occasion for regenerating or re-vitalizing the categories that operate in culture, *and* for occasions during which people clash with those categories, leaving them trapped in a condition of limbo and relegated among those who share those alien characteristics. Without adequate ways of ‘devising’ people’s ‘spontaneous’ experiences of transition, liminality can reduce to something disquieting and disturbing which undermines language, prevents us from naming things and crumbles clichés and the canonical relationship between words and things [1].

## 4. Conclusions and Future Prospects

We have provided a theoretically informed critical review of liminality literature in the field of oncology informed by process thinking, as outlined in Section 2. This literature makes extensive use of qualitative methodologies (interpretative phenomenological analysis, thematic analysis, content analysis, grounded theory, etc.) using material collected through semi-structured interviews, narratives, stories of life, testimonies and field experiences of a socio-ethnographic type. This allows a high level of sensitivity to the subjective and cultural dimensions of illness experiences, and uses most of the general conceptual categories present in liminality theory, including:-Ritualized ceremonies symbolizing the beginning, middle and end points of becomings;-The replacement of normative structure with a creative anti-structure;-Changed perception of time and space;-Generation of *communitas* shared between liminars;-A process of discussion, negotiation and the building of new identities.

The notion of liminality has a great potential for innovating research devoted to understanding the situated experience of patients from a subjective point of view in the different phases of the experience of illness from receipt of diagnosis to palliative care. There is a need, however, not just for more empirical research, but for greater attention to the theory informing it. This is necessary because liminality is a theoretical concept whose complexity is differently comprehended and ‘nuanced’ by different authors. The review offered differs from a Qualitative Evidence Synthesis by focusing as much upon the theoretical framing of the research as upon a survey and synthesis of findings.

Starting with the seminal contribution from Little et al., in 1998 [1], we identified an *existentialization* of liminality theory with subtle shaping effects upon the conclusions drawn about acute and enduring phases of liminality following diagnosis. There is a risk here of oversimplifying the notion of liminality, transforming it into a ready-to-use and pre-defined *a priori* label. A stream of subsequent work has followed in this existentialist tradition, but other publications have indicated its limitations. Several studies advocated broadening the perspective to include the ‘generative potential’ associated with liminal experience. This generative potential centers on the capacity of liminal experience to foster new *becomings* supported by new modes of meaning making capable of addressing life ruptures and crafting new ways of meaningfully ‘going on’ [43,80,86]. Others advocated accommodating more of the social surround into the picture of illness. For example, liminality research has proved relevant not just to patients but also to their relatives and carers, and beyond this, to the ever-changing institutional and societal settings within which health care unfolds as a historical form of life. In particular, the review surfaced a critique of the tendency to treat liminality as a stable *psychological state* operative, following an existential limit experience, at the *level of the individual*. The two theoretical broadenings noted above question these assumptions and point towards liminality as a *process of transition* that is ultimately irreducible to the individual level (though most certainly relevant at a personal level).

A further tension was identified between the existential interpretation of liminality and a process theoretical interpretation. The process orientation of van Gennep and Turner emphasizes the paradoxical qualities of liminal experience, as summed up in the formulation of both/and as well as neither/nor. As highlighted in Section 3.7, much of the surveyed literature draws attention to this experience of paradox, which results from the ‘occupancy’ of the (non-) position of transition between a form of life that is ‘no longer’ and a form that is ‘not yet’. A great value of liminality theory from a processual perspective lies in its capacity to recognize the reality of unstable experiences of transition that otherwise might be disqualified as illogical hence unreal. The clarity of the existential framing, by contrast, comes at the cost of reducing (‘de-paradoxifying’) the paradox to one of its mutually contradictory terms. For example, the affective ambivalence of liminal experience that is crucial from a processual perspective is reduced to sheer negativity when existentialized as pure ‘dread’ and ‘angst’. What this de-paradoxification obscures is precisely the *generative* capacity proper to the paradoxes and ambiguities of liminal experience. It is this generativity in the face of paradox that shows up as fundamental to the subgroup of literature dealing with ritual and art therapeutic interventions into the patient experience dealt with in Section 3.6. In ‘devised’ liminal situations mediated by these ‘liminal technologies’, this paradox and ambivalence is expected, contained and worked through in a safe space.

In what remains, we offer some suggestions about future directions for liminality research in oncology. Greater attention should be given to five related issues: the specificity and temporality of the experience of emergence (*which* emergency provokes liminality?); the management and patterning of that experience by means of symbolic mediation (*how* is the emergency ‘interpreted’); the accent on becoming (transition to *what*?); the subsequent importance of chronicity and hence time experience (questions of *when*?); and the embeddedness of patient experience within cultural and social systems which are themselves describable in liminality-theoretic terms.

First, the experience of emergence concerns the connection of liminal experience to the *emergence of novelty* associated with the disruptions to ‘forms of life’ that are occasioned by the advent of a ‘newcomer’ of some sort (e.g., the emergency provoked by the arrival of cancer, often via its diagnosis, or by a particular treatment regime). Attention to this newcomer and its impact on established forms of sensemaking is crucial, but for the most part, the existing liminality literature lacks adequate accounts of emergence.

Second, greater recognition must be given to the fact that ‘liminal experience’ is always subject to dynamics of culturally mediated *interpretation* (or ‘sensemaking’) by means of symbolic forms of different kinds. Lacking this focus, the existing literature risks ‘essentializing’ liminal experience as if it were the natural and inevitable outcome of encounters with cancer, and missing the multiple ways in which liminal experience can be symbolically formed via semiotic elaboration of various kinds (e.g., it can be formed as ‘sacred’ by the mediation of ritual, as ‘aesthetic’ by the mediation of art, etc.).

Third, in both the classic and recent literature on liminal experience, the semiotic elaboration of liminal experience by means of these symbolic forms is geared towards the *problem of becoming other*. Here, we find the distinction between ‘spontaneous’ and ‘devised’ liminal experiences particularly fruitful and in need of further development. Applications within the health field tend to be geared less to becoming other than to the problematic of *returning* to health, and hence, restoring a lost normality. Researchers who neglect this issue risk getting ‘stuck’ on the question of whether illness experiences necessarily involve a transformative dynamic of becoming other.

Fourth, this emphasis on novel emergence, symbolic mediation, and becoming foregrounds the relevance of the temporal dimension of chronicity. To date, applications in the health field have treated this temporal dimension only superficially, and have preferred to emphasize the more obvious but perhaps less profound notion of liminal space (see Jellema et al. for a more sophisticated treatment of the spatial dimensions of liminality in cancer care [2]). Grasped within a process ontology, liminality theory insists that space and time cannot be separated without distortion [21].

Finally, liminality theory should encourage a broadening of scope to accommodate greater historical and societal context into oncology research into patient experience. An interesting point raised by Little et al. [1] is that there is something distinctively modern about cancer experiences that have increased precisely because so many other ways of getting sick have been eliminated or largely contained. We now expect to return to health after disease, whereas for most of human history, chronic disease is likely to have been the norm, not the exception, coupled with the fact that people died much younger. Once a familiar and expected part of ordinary life [87,88], now, members of wealthy modern societies are shocked by serious illness and death. In this context, liminality theory carries potential for revising the notion of “chronicity” in health [89] beyond the well-established distinction between ‘chronic’ and ‘acute’ types of disease. We noted that liminality research has focused on chronic diseases of two main types, of which cancer is one paradigm and conditions whose explanation remains troubled is another. Both types of chronicity are dramatically on the rise as a function of increased life expectancy, early detection techniques and therapeutic successes based on scientific research. Furthermore, in today’s health system, one finds a growing focus on patient empowerment, participatory decision-making processes (beyond the classic position of compliance and adherence) and ‘responsibilization’ of patients for their own care processes. This requires the acquisition of skills to deal with the evolution of one’s health conditions, and the search for future plans and new projects of well-being and quality of life that are connected to one’s contexts of belonging. Liminality theory adequately understood holds potential to attend to the processual dimension of the experience of illness in the field of oncology and in that of health psychology in general, offering a reading of the disease phenomena capable of holding together both the psychological-subjective and the social-normative aspects. For example, conceived as a core concept of transdisciplinary process thought [21], liminality provides a conceptual tool for ‘zooming out’ and reflecting upon the bigger picture of how health psychology is itself implicated in a societal regime of truth which must regulate the liminal zone between the experiential knowledge of the patient perspective and the experimental knowledge proper to evidence-based medicine. This could illuminate tensions that exist in the broader context of technogoverance [90] between “evidence-based medicine” and “patient-centered medicine” [91,92,93,94,95,96,97,98], a situation that can itself be grasped as a liminal predicament that generates liminal experiences throughout the system. To date, however, the field of liminality-theoretic research has concentrated almost exclusively on the alleged liminality of the patient experience with no mention of how such re-description of ‘patient experience’ is itself part of the contemporary technogovernance of health.

Taken together, these five foci would enable the development of a more thoroughgoing process ontological framing for research on liminality. It would help to counteract more reductive uses of the concept of liminality and stimulate a coherent body of research capable of ‘capturing’ the less tangible transitional and experiential dimensions of living with cancer [7,9,21,34,99,100,101].

## Data Availability

Not applicable.

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
