# Peer review of "A Theoretically Informed Critical Review of Research Applying the Concept of Liminality to Understand Experiences with Cancer: Implications for a New Oncological Agenda in Health Psychology"

_ijerph, 2023, doi:10.3390/ijerph20115982_

Round 1

Reviewer 1 Report

The review proposal is interesting for health psychologists, particularly in the field of psycho-oncology. Introduces a complete state of art.

We recommend generating a timeline scheme about the beginning of liminality, and the present and future direction where the authors intend to take the reader. It does not necessarily have to be linear, it can be a robust tree of branches on the proposed agenda.

In point 6.1, the reference is incomplete.

It is also convenient to highlight, it can be in a Table 3 axes that improve your proposal:

1. What is known about the subject, for example, the relevant points of section 4: Name | Description.

2. What is not known about the subject, highlighting the shortcomings of the current models, what they have not covered, what is not clear, or lack of qualitative/quantitative studies.

3. This proposal. What the authors want to emphasize, is point out the guidelines where they want to take the research, with replicable or adjustable methods for each psycho-oncology group.

Author Response

response is attached

Reviewer 2 Report

13.04.2023r.

Oncology, chronicity and liminality: a meta-theoretical review on the state of affair and implications for a new agenda in health psychology

The reviewed manuscript titled Oncology, chronicity and liminality: a meta-theoretical review on the state of affair and implications for a new agenda in health psychology is an interesting review.

The authors made a good review of the literature and highlighted the value of liminality as a process of concrescence.

However:

The work, although it is undoubtedly the result of the authors' efforts, is very long-winded, and therefore - in my opinion - not very legible. Individual chapters seem to be vague. A few more specifics and references can be found only in chapter 6, but the previous parts, unfortunately, do not encourage the reader to read the rest of the article. I suggest significantly shortening the article, especially chapters 1-5.

In addition, a minor spell check is required.

Despite some reservations, the subject matter of the article is very interesting, and the Authors' commitment is evident. I suggest condensing the article and resubmitting it for re-evaluation (after taking into account the comments of the reviewer).

Author Response

response is attached

Reviewer 3 Report

Dear authors,

I found your investigation on this topic to be interesting, as it is an area that is underexplored in the literature. However, I believe that this article does not align with the aims and scope of the present journal. Although the meta-theoretical review is very fascinating, it represents a philosophical and anthropological contribution (e.g., there are many references to Greek myths with few scientific and clinical repercussions), describing an existentialist point of view of the oncological experience without any specific clinical implications. 

Moreover, the methodology used for selecting the articles has not been explained in the paper, which is a fundamental requirement for any scientific publication. Therefore, I suggest reporting the methodology used to identify the literature for this review.

To enhance the scientific rigor of the paper and align it with the objectives of the present journal, I recommend conducting a meta-theoretical systematic review or including a meta-synthesis of the qualitative studies that have studied the concept of 'liminality' in oncological populations. This will add weight to the findings and conclusions of the paper and help it stand out as a valuable contribution to the field.

Best wishes

Author Response

response is attached

Round 2

Reviewer 2 Report

19.05.2023

Second review of the article

"A theoretically informed critical review of research applying the concept of liminality to understand experiences with cancer: implications for a new oncological agenda in health psychology"

The Authors took into account the reviewer's comments. Doubts remain about the philosophical contribution - I'm not sure if such an article profile is appropriate for this journal. At the same time, I appreciate the Authors' contribution and efforts related to the implementation of the topic. Despite some doubts, I believe that the publication of this unconventional article is worth considering, but after taking into account all the reviews.

Author Response

Comments are in the attached letter

Reviewer 3 Report

Dear authors,

I appreciate your effort in reviewing the paper. I found your proposal of a new framework for clinical and health psychology very interesting.

However, I still believe the manuscript is very long and redundant, I would suggest to provide a shorter and more concise version and clarifying inclusion and exclusion criteria for the articles selected.

In addition, I believe it is not necessary to refer to the special issue of the journal in the introduction of the paper.

Best regards

Author Response

Comments are in the attached letter

Round 3

Reviewer 3 Report

Dear authors,

The revised version is now much more concise and accessible, making it easier for both clinicians and researchers to comprehend the key points and findings of your research. I appreciate your efforts in improving the manuscript and I endorse its publication. 

I noticed that there may be some inconsistencies in the numbering of the paragraphs. Please, carefully review it and ensure the accurate numbering of all paragraphs throughout the article.

Best regards